# Coumarins from *Seseli devenyense* Simonk.: Isolation by Liquid–Liquid Chromatography and Potential Anxiolytic Activity Using an In Vivo Zebrafish Larvae Model

**DOI:** 10.3390/ijms22041829

**Published:** 2021-02-12

**Authors:** Jarosław Widelski, Simon Vlad Luca, Adrianna Skiba, Monika Maciąg, Barbara Budzyńska, Laurence Marcourt, Jean-Luc Wolfender, Krystyna Skalicka-Woźniak

**Affiliations:** 1Department of Pharmacognosy with Medicinal Plant Unit, Medical University of Lublin, 20-093 Lublin, Poland; 2Biothermodynamics, TUM School of Life Sciences, Technical University of Munich, 85354 Freising, Germany; vlad.luca@tum.de; 3Department of Pharmacognosy, Grigore T. Popa University of Medicine and Pharmacy Iasi, 700115 Iasi, Romania; 4Independent Laboratory of Natural Products Chemistry, Medical University of Lublin, 20-093 Lublin, Poland; adrianna.skiba@umlub.pl (A.S.); kskalicka@pharmacognosy.org (K.S.-W.); 5Independent Laboratory of Behavioral Studies, Medical University of Lublin, 20-093 Lublin, Poland; monika.maciag@umlub.pl (M.M.); barbara.budzynska@umlub.pl (B.B.); 6Department of Biopharmacy, Medical University of Lublin, 20-093 Lublin, Poland; 7School of Pharmaceutical Sciences, Institute of Pharmaceutical Sciences of Western Switzerland, IPSWS, University of Geneva, CMU, 1211 Geneva, Switzerland; laurence.marcourt@unige.ch (L.M.); Jean-Luc.Wolfender@unige.ch (J.-L.W.)

**Keywords:** *Seseli devenyense*, Apiaceae, countercurrent chromatography, central nervous system, anxiety, *Danio rerio*

## Abstract

Different types of anxiety disorders have become the number one mental health issue in developed countries. The search for new, safer and effective drug-like molecules among naturally derived substances faces two difficulties: an efficient method of isolation compounds with a high-purity and high-throughput animal model for activity assay. Thus, the aim of the present study was to isolate by liquid–liquid chromatography high-purity rare coumarins from the fruits of *Seseli devenyense* Simonk. and evaluate their anxiolytic effect (defined as reversed thimotaxis) using a 5-days post-fertilization (dpf) *Danio rerio* larvae model. Liquid–liquid chromatography enabled the isolation of one simple hydroxycoumarin (devenyol) and four pyranocoumarins (cis-khellactone, d-laserpitin, isolaserpitin and octanoyllomatin). The anxiolytic effect was defined as a decrease in the time spent in the boundaries of the living space (also described as reversed thigmotaxis). Our results show that all isolated courmarins exerted a significant influence on the anxiety behavior (anxiolytic activity) in the zebrafish larvae model. According to our knowledge, this is the first report of anxiolytic activity of pyranocoumarins and devenyol.

## 1. Introduction

Anxiety is a complex mental disorder characterized by excessive worries, irrational fear and/or avoidance of objects, places and situations. Anxiety disorders can be classified as: general anxiety disorder (GAD), panic disorders, phobias, social anxiety disorders, obsessive-compulsive disorders (OCD) and post-traumatic stress disorder (PTSD) [1]. According to the World Health Organization (WHO), the total number of people suffering from anxiety is estimated at over 264 million [2].

Despite the fact that the precise pathophysiology of anxiety is still not fully elucidated, one of the most widely accepted theories points to disturbances in the gamma-aminobutyric acid (GABA)-ergic system [3]. Gamma-aminobutyric acid (GABA) is one of the most important inhibitory neurotransmitters in the central nervous system (CNS). Its receptors are localized in the brain, as well as in peripheral nerves [4]. Experimental and clinical studies have proven that activation of GABA-ergic neurotransmission is responsible for decreasing anxiety level (anxiolytic activity) [5,6]. A great number of clinically approved drugs, such as benzodiazepines, barbiturates, gabapentin, pregabalin, valproate, vigabatrin or tiagabine exert anxiolytic activity by modulation of the GABA-ergic system [7]. Although generally considered as safe and effective, the currently available anxiolytic therapy is responsible for numerous side effects, such as decrease in psychomotor activity, problems with verbal learning, disturbances of memory processes and pharmacological tolerance [8]. Therefore, there is an imperious demand for discovering new groups of pharmaceuticals with an improved safety profile.

Due to the huge structural heterogeneity and great number of allosteric binding sites, many natural compounds were discovered to possess GABA_A_ receptor modulating activity [7]. Having the adequate liphophilicity and size to cross the blood–brain barrier, naturally occurring coumarins have recently gained an increasing popularity as potential drug scaffolds with CNS modulatory activities (e.g., anticonvulsant, antineurodegenerative, procognitive, neuroprotective or antidepressant) [9].

Comprised of around 135 species, of which 55 distributed over Central and Eastern Europe, genus *Seseli* L. (Apiaceae) is known to be a very rich source of diverse coumarins [10]. Some representatives of *Seseli* genus have been used in the treatment of CNS disorders. According to ethnopharmacological studies, *Seseli tortuosum* L. was used in the traditional medicine of Iran to treat epilepsy [11]. A volatile fraction obtained from the aerial parts of *Seseli sibiricum* (L.) Garcke potentiated the action of pentobarbital in rats, and generally acted as a tranquilizer, but had no anticonvulsant activity [12]. The ensemble of essential oils and extracts of *Seseli rigidum* Waldst. & Kit. exhibited remarkable inhibitory activity toward acetyltcholinesterase (AChE) and butyrylcholinesterase (BChE) [13], which may implicate an increase of the CNS levels of acetylcholine with subsequent improvement of cognitive processes.

*Seseli deveyense* Simonk. was previously reported to possess rare coumarins (e.g., devenyol, deveniosides A–C) or unique coumarin structures, such as the decanoic and dodecanoic esters of lomatin and cis-khellactone [14]. However, to the best of our knowledge, the pharmacological activity of courmarins from *Seseli devenyense* was not previously evaluated, especially at CNS level.

Zebrafish (*Danio rerio*) shows a 70% gene homology with mammals, including humans [15]. Additionally, a high resemblance of the neurotransmitter signaling pathways, such as cholinergic, serotoninergic, dopaminergic and noradrenergic systems, was observed [16]. Consisting of diencephalon, telencephalon, cerebellum, peripheral nervous system with sensory and motor neurons, the zebrafish brain structures exhibit a high similarity to those observed in vertebrates [16,17]. These similarities, together with low-cost and time-sparing experiments, make zebrafish a perfect in vivo model for CNS screening assays. Furthermore, the wide range of complex mammalian-like behaviors, the easy-to-observe behavioral endpoints and endophenotypes make the zebrafish animal model an effective screening tool for evaluation of anxiety-like behaviors.

The aim of the study was to isolate high-purity rare coumarin derivatives present in the fruits of *Seseli devenyense* Simonkai. using liquid–liquid chromatography and determine their potential anxiolytic activity in an in vivo zebrafish larvae (5 days post fertilization, dpf) model. According to our knowledge, this is the first report of anxiolytic activity of pyranocoumarins.

## 2. Results

### 2.1. Liquid–Liquid Chromatography Isolation of Coumarins

The chromatogram obtained as results of preliminary HPLC-DAD analysis of the *Seseli devenyense* methanolic fruit extract (SDMFE) displayced the presence of several coumarin compounds. Their chemical character and structure were confirmed by HPLC-ESI-QTOF-MS/MS (Appendix A). The partition coefficient (*P_i_*) values obtained from the shake-flask experiments were used as screening parameter for selecting the biphasic solvent systems that could allow the liquid–liquid chromatography separation of coumarins from the crude extract. Various hexane/ethyl acetate/methanol/water (HEMWat) mixtures were tested (Table 1).

Considering that for a predictable and optimum separation the *P_i_* values should be situated between 0.4 and 2.5 [18], the following biphasic solvent systems were selected for the subsequent liquid–liquid chromatography experiments: *solvent system I* HEMWat 2/5/2/5 (*v*/*v*/*v*/*v*) for the separation of coumarins **1** and **2**; *solvent system II* HEMWat 6/5/6/5 (*v*/*v*/*v*/*v*) for the separation of coumarins **3** and **4**; and *solvent system III* HEMWat 5/2/5/2 (*v*/*v*/*v*/*v*) for the separation of coumarin **5**. Parallel batch liquid–liquid chromatography experiments with the three solvent systems were performed under identical operating conditions (Appendix A). Based on the total separation times (~3.5 h) of the three liquid–liquid chromatography experiments, the total mobile phase consumption was ~1.3 L (Table 2). The column preparation times (column filling, column equilibration, column cleaning) were not included in these calculations.

After analyzing by HPLC-DAD all collected fractions, those containing high-purity coumarins (>95.0%) were pooled together. Compound **1** (5.5 mg, 97.5%), compound **2** (9.2 mg, 99.3%), compound **3** (1.4 mg, 98.5%), compound **4** (1.4 mg, 96.5%) and compound **5** (3.1 mg, 99.6%) were thus obtained (Appendix A). The identity of the purified coumarins, confirmed by HPLC-ESI-QTOF-MS/MS, 1D- and 2D-NMR, was established as follows: devenyol (**1**), cis-khellactone (**2**), d-laserpitin (**3**), isolaserpitin (**4**) and octanoyllomatin (**5**) (Figure 1). The spectroscopic data (Appendix A) were in agreement with those available in spectral libraries and literature (14, 19, 20). Next, the yields of the isolated compounds obtained by liquid–liquid chromatography were compared with those obtained by conventional solid–liquid chromatography (open column chromatography and preparative HPLC) [14]. It was noticed that the liquid–liquid chromatography yields were 5-fold higher for devenyol (**1**), 17-fold higher for cis-khellactone (**2**), 4.6-fold higher for d-laserpitin (**3**), 3-fold higher for isolaserpitin (**4**) and 1.6-fold higher for octanoyllomatin (**5**) (Table 2).

### 2.2. The Effects of Seseli devenyense Crude Extract and Isolated Coumarins on Thigmotaxis Behaviours of the Zebrafish Larvae during Light-Dark Changes

#### 2.2.1. The Effects of SDMFE Extract on Thigmotaxis Behaviors of the Zebrafish Larvae during Light-Dark Changes

For a better characterization of the effect of SDMFE on the locomotor activity of the *Danio rerio* larvae in response to a light–dark challenge, the average distance traveled per minute in all three light–dark cycles was determined (Figure 2a). Two-way ANOVA showed statistically significant changes in the light–dark condition response [F(1, 448) = 3.60, *p* = 0.0587], treatment effect [F(5, 448) = 29.46, *p* < 0.001], as well as interaction [F(5, 448) = 6.92, *p* < 0.0001]. The post hoc Bonferroni’s test showed an increase in locomotor activity of control group (DMSO-treated, *p* < 0.001) during the dark phase in comparison to the light phase. During the dark phase, a significant decrease in the locomotor activity was observed when the 5-dpf larvae of *Danio rerio* were treated with diazepam (*p* < 0.01) and SDMFE at 9 and 12.5 μg/mL (*p* < 0.05) and 25 μg/mL (*p* < 0.001), in comparison with the DMSO-treated group in the dark phase (Figure 2a).

The influence of SDMFE treatment during the light–dark transitions on the thigmotactic behaviors (when the distance traveled in the central arena was considered) of the zebrafish larvae was also studied (two-way ANOVA light–dark condition [F(1, 448) = 192.37, *p* < 0.0001), treatment [F(7, 448) = 61.14, *p* < 0.0001] and interaction [F(7, 448) = 25.24, *p* < 0.0001]). The post hoc test (Bonferroni’s test) showed an increase in the percentage of the distance traveled by the larvae in the central area in the control group (*p* < 0.05), diazepam (*p* < 0.001) and after incubation in SDMFE at 3, 9 and 12.5 μg/mL (*p* < 0.05) as well as at 25 μg/mL (*p* < 0.01) in the dark phase, as compared with the light phase. In addition, during the dark phase of the experiment, a significant increase in the locomotor activity in the central area was observed when 5-dpf larvae were treated with diazepam (*p* < 0.001) and SDMFE at 9 (*p* < 0.05) and 25 μg/mL (*p* < 0.01), as compared with the DMSO-treated group in the dark phase (Figure 2b). During the light phase of the experiment, a significant increase in the percentage of the distance moved in the central arena was noticed when 5-dpf larvae were treated with diazepam (*p* < 0.05) and SDMFE at 25 μg/mL (*p* < 0.05), when compared to the DMSO-treated group in the light phase (Figure 2b).

In Figure 2c, the effect of different concentrations of SDMFE on the thigmotaxis of the larvae is presented; the duration of the activities in the central area was measured (two-way ANOVA light–dark condition [F(1, 448) = 71.31, *p* < 0.001], interaction [F(6, 448) = 36.58, *p* < 0.0001) and treatment effect [F(6, 448) = 33.44, *p* < 0.001]). The post hoc test (Bonferroni’s test) showed an increase in the percentage of the time spent by the larvae in the central area in the control group and diazepam (*p* < 0.001), as well as in the group after incubation in SDMFE at 3, 6, 9 (*p* < 0.05) and 12.5 μg/mL (*p* < 0.01) in the dark phase, as compared with the light phase. In addition, during the dark phase of the experiment, a significant increase in the locomotor activity in the central area were observed when 5-dpf larvae were treated with diazepam (*p* < 0.001) and a decrease with SDMFE at 25 μg/mL (*p* < 0.01), as compared with the DMSO-treated group in the dark phase (Figure 2c). During the light phase of the experiment, a significant increase in the percentage of the distance moved in the central arena was noticed when 5-dpf larvae were treated with diazepam (*p* < 0.05), as compared to the DMSO-treated group in the light phase (Figure 2c).

#### 2.2.2. The Effects of Devenyol on Locomotor Activity and Thigmotaxis Behaviors of the Zebrafish Larvae during Light-Dark Changes

Two-way ANOVA showed statistically significant changes of the zebrafish larvae behavior in the light–dark condition response [F(1, 427) = 441.07, *p* < 0.0001], treatment effect [F(6, 427) = 32.72, *p* < 0,0001], as well as interaction [F(6, 427) = 4.64, *p* = 0.0001]. The post hoc Bonferroni’s test showed decreases in the locomotor activity during the dark phase after incubation with diazepam (*p* < 0.001) and devenyol at 1.5 μM (*p* < 0.05), in comparison with the DMSO-treated group in the dark phase (Figure 3a). In addition, in the dark challenge phase of the experiment, significant increases related to the locomotor activity were observed when the 5-dpf larvae were treated with devenyol at concentrations of 1.5, 3, 6, 9 and 15 μM (*p* < 0.001), in comparison to the light phase.

The influence of devenyol on the light–dark transitions was also studied regarding the thigmotaxis behavior. In Figure 3b, it can be seen that the treatment with devenyol in the dark challenge influenced the thigmotaxis behavior of the larvae, when the covered distance was considered (two-way ANOVA light–dark condition [F(1, 427) = 320.98, *p* < 0.0001), treatment [F(6, 427) = 33.17, *p* < 0.0001] and interaction [F(6, 427) = 6.23, *p* < 0.0001]). The post hoc test (Bonferroni’s test) showed an increase in the percentage of the distance travelled by the larvae in the central area after incubation in diazepam (*p* < 0.001) and devenyol at 1.5, 3, 6, 9, and 15 μM (*p* < 0.001) in the dark phase, as compared with the light phase. In addition, during the dark phase of the experiment, significant increases in the locomotor activity in the central area were observed when 5-dpf larvae were treated with DMSO (*p* < 0.01), diazepam (*p* < 0.001) and devenyol at concentrations of 1.5, 9 and 15 μM (*p* < 0.05) and 6 μM (*p* < 0.01), as compared with the DMSO-treated group in the dark phase (Figure 3b). During the light phase of the experiment, a significant increase in the percentage of the distance moved in the central arena was noticed when 5-dpf larvae were treated with diazepam (*p* < 0.001), as compared to the DMSO-treated group in the light phase (Figure 3b).

In Figure 3c, the effect of different concentrations of devenyol on thigmotaxis of the larvae is presented; the duration of the activities in the central area was measured (two-way ANOVA light–dark condition [F(1, 427) = 379.15, *p* < 0.001], interaction [F(1, 427) = 2.67, *p* < 0.0147) and treatment effect [F(6, 427) = 41.47 *p* < 0.001]). The post hoc Bonferroni’s test showed an increase in the percentage of the time spent in the central arena during the dark phase after DMSO-treatment (*p* < 0.01), diazepam (*p* < 0.001), devenyol at all concentrations (*p* < 0.001), as compared to the light phase. During the dark challenge phase, a significant increase in this parameter was observed after treating the 5 dpf larvae with diazepam (*p* < 0.001) and devenyol at concentrations of 1.5, 6, 9, and 15 μM (*p* < 0.05), as compared with the DMSO-treated group in the dark phase (Figure 3c). During the light phase of the experiment, a significant increase in the percentage of the time spent in the central arena was noticed when 5-dpf larvae were treated with diazepam (*p* < 0.001), as compared to the DMSO-treated group in the light phase (Figure 3c).

#### 2.2.3. The Effects of cis-Khellactone on Locomotor Activity and Thigmotaxis Behaviors of the Zebrafish Larvae during Light-Dark Changes

Two-way ANOVA showed statistically significant changes in light–dark condition response [F(1, 512) = 15.34, *p* = 0.0001], treatment effect [F(7, 512) = 51.90, *p* < 0.0001], as well as interaction [F(7, 512) = 6.615, *p* < 0.0001]. The post hoc Bonferroni’s test showed increase in the locomotor activity during the dark phase in control group (DMSO, *p* < 0.01) and cis-khellactone at concentrations of 1.5 μM (*p* < 0.001), 3 μM (*p* < 0.01), 6 μM (*p* < 0.01), 9 μM (*p* < 0.01), 12.5 μM (*p* < 0.001) and 25 μM (*p* < 0.01), in comparison with the light phase (Figure 4a). During the dark phase, significant changes in the locomotor activity were observed after diazepam administration (*p* < 0.01), when compared to DMSO-treated group in the dark phase (Figure 4a). During the light phase of the experiment, significant changes in the locomotor activity were not noticed, when compared to DMSO-treated group in light phase (Figure 4a).

The light–dark transitions were also studied for thigmotaxis behavior on the zebrafish larvae. In Figure 4b, the total distance moved by zebrafish larvae in the central area in each minute was provided. For zebrafish larvae at 5-dpf, the treatment with different concentration of cis-khellactone influenced the thigmotaxis behaviors of the larvae, when considering the moved distances (two-way ANOVA light–dark condition [F(1, 512) = 312.3, *p* < 0.0001), treatment [F(7, 512) = 11.93, *p* < 0.0001] and interaction [F(7, 512) = 29.48, *p* < 0.0001]). The post hoc test (Bonferroni’s test) showed an increase in the percentage of the distance traveled by the larvae in the central area in the control group (DMSO, *p* < 0.01), diazepam (*p* < 0.001) and cis-khellactone at 1.5, 9 and 12.5 μM (*p* < 0.001), as well as at 6 and 25 μM (*p* < 0.01) in the dark phase, as compared with the light phase. In addition, during the dark phase of the experiment, significant changes in the distance traveled in the central area were not observed, as compared with the DMSO-treated group in the dark phase (Figure 4b). During the light phase of the experiment, a significant increase in the percentage of the distance moved in the central arena was noticed when 5-dpf larvae were treated with diazepam (*p* < 0.05), as compared to the DMSO-treated group in the light phase (Figure 4b).

In Figure 4c, the effect of different concentrations of cis-khellactone on thigmotaxis of the larvae is presented. The duration of the activities in the central area was measured (two-way ANOVA light–dark condition [F(1, 512) = 174.2, *p* < 0.0001], interaction [F(7, 512) = 20.71, *p* < 0.0001) and treatment effect [F(7, 512) = 6.847, *p* < 0.0001]). The post hoc test (Bonferroni’s test) showed an increase in the percentage of the time spent by the larvae in the central arena in the control group DMSO (*p* < 0.05), diazepam (*p* < 0.001) and cis-khellactone at concentrations of 1.5, 3, 12.5, 25 μM (*p* < 0.001) in the dark phase, as compared with the light phase. In addition, during the dark phase of the experiment, significant increases in the time spent in the central arena were observed when 5-dpf larvae were treated with diazepam (*p* < 0.01) and cis-khellactone in concentrations of 1.5 μM (*p* < 0.01) and 3, 9, 12.5 and 25 μM (*p* < 0.001), as compared with the DMSO-treated group in the dark phase (Figure 4c). During the light phase of the experiment, a significant increase in the percentage of the time spent in the central arena was noticed when 5-dpf larvae were treated with diazepam (*p* < 0.05) and cis-khellactone at 6 μM (*p* < 0.05) and 9 μM (*p* < 0.001), as compared to DMSO-treated group in the light phase (Figure 4c).

#### 2.2.4. The Effects of D-Laserpitin on Locomotor Activity and Thigmotaxis Behaviors of the Zebrafish Larvae during Light-Dark Changes

Two-way ANOVA showed statistically significant changes in the light–dark condition response [F(1, 320) = 15.34, *p* = 0.0001], treatment effect [F(7, 320) = 51.90, *p* < 0.0001], as well as in the interaction [F(7, 320) = 6.615, *p* < 0.0001]. The post hoc Bonferroni’s test showed an increase in the locomotor activity during the dark phase in the control group (DMSO, *p* < 0.001) and d-laserpitin at concentrations of 25 μM (*p* < 0.01) and 50 μM (*p* < 0.001), in comparison with the light phase (Figure 5a). During the dark phase, significant decreases in the locomotor activity were observed when the larvae were treated with diazepam (*p* < 0.01) and d-laserpitin at concentrations of 12.5 μM (*p* < 0.001), 25 μM (*p* < 0.001) and 50 μM (*p* < 0.01), when compared to the DMSO-treated group in the dark phase (Figure 5a).

The light–dark transitions were also studied for thigmotaxis behavior on the zebrafish larvae. The treatment with different concentrations of d-laserpitin influenced the thigmotaxis behaviors of the larvae, when evaluating the moved distances (two-way ANOVA light–dark condition [F(1, 320) = 312.3, *p* < 0.0001), treatment [F(7, 320) = 11.93, *p* < 0.0001] and interaction [F(7, 320) = 29.48, *p* < 0.0001]) (Figure 5b). The post hoc test (Bonferroni’s test) showed an increase in the percentage of the distance traveled by larvae in the central area in control group (DMSO, *p* < 0.001), diazepam (*p* < 0.001) and d-laserpitin at all concentrations (12.5 and 25 μM (*p* < 0.001) and 50 μM (*p* < 0.01) in the dark phase, as compared with the light phase. In addition, during the dark phase of the experiment, significant increases in the distance traveled in the central area were observed when 5-dpf larvae were treated with diazepam (*p* < 0.001) and d-laserpitin at 12.5, 25 and 50 μM (*p* < 0.001), as compared with the DMSO-treated group in the dark phase (Figure 5b). During the light phase of the experiment, a significant increase in the percentage of the distance moved in the central arena was noticed when 5-dpf larvae were treated with diazepam (*p* < 0.05) and d-laserpitin 25 μM (*p* = 0.05) and 50 μM (*p* < 0.001), as compared to the DMSO-treated group in the light phase (Figure 5b). The concentration of 12.5 μM decreased (*p* < 0.01) the percentage of the distance moved in the central arena in light phase, in comparison to the DMSO-treated group in the light phase (Figure 5b).

In Figure 5c, the effect of different concentrations of d-laserpitin on thigmotaxis of the larvae is presented; the duration of the activities in the central area was measured (two-way ANOVA light–dark condition [F(1, 320) = 174.2, *p* < 0.0001], interaction [F(7, 320) = 20.71, *p* < 0.0001) and treatment effect [F(7, 320) = 6.847, *p* < 0.0001]). The post hoc test (Bonferroni’s test) showed an increase in the percentage of the time spent by the larvae in the central arena in the control group (DMSO, *p* < 0.01), diazepam (*p* < 0.001) and d-laserpitin at concentrations of 25 μM (*p* < 0.001) and 50 μM (*p* < 0.01) in the dark phase, as compared with the light phase. In addition, during the dark phase of the experiment, significant increases in the time spent in the central arena were observed when 5-dpf larvae were treated with diazepam (*p* < 0.05) and d-laserpitin at concentrations of 12.5, 25 and 50 μM (*p* < 0.001), compared with the DMSO-treated group in the dark phase (Figure 5c). During the light phase of the experiment, a significant increase in the percentage of the time spent in the central arena was noticed when 5-dpf larvae were treated with diazepam (*p* < 0.05) and d-laserpitin at 12.5 and 50 μM (*p* < 0.001), as well as 25 μM (*p* < 0.01), when compared to the DMSO-treated group in the light phase (Figure 5c).

#### 2.2.5. The Effects of Isolasepitin on Locomotor Activity and Thigmotaxis Behaviors of the Zebrafish Larvae during Light-Dark Changes

Two-way ANOVA showed statistically significant changes in the light–dark condition response [F(1, 320) = 53.97, *p* < 0.0001], treatment effect [F(4, 320) = 272.31, *p* < 0.0001], as well as interaction [F(4, 320) = 10.70, *p* < 0.0001]. The post hoc Boniferroni’s test showed an increase in the locomotor activity during the dark phase only in the control group (DMSO, *p* < 0.01), in comparison with the light phase (Figure 6a). During the dark phase, significant increases in the locomotor activity were observed when the larvae were treated with isolaserpitin at concentrations of 12.5, 25 and 50 μM (*p* < 0.001) and a decrease in regard to diazepam (*p* < 0.01), when compared to DMSO-treated group in the dark phase (Figure 6a).

During the light phase of the experiment, a significant increase in the locomotor activity was noticed when 5-dpf larvae were treated with isolaserpitin at 12.5, 25 and 50 μM (*p* < 0.001), when compared to the DMSO-treated group in the light phase (Figure 6b).

The light–dark transitions were also studied for thigmotaxis behavior on the zebrafish larvae. In Figure 6b, isolaserpitin influenced the thigmotaxis behaviors of the larvae, when the moved distance was considered (two-way ANOVA light–dark condition [F(1, 320) = 236.24, *p* < 0.0001), treatment [F(4, 320) = 90.48, *p* < 0.0001] and interaction [F(4, 320) = 23.08, *p* < 0.0001). Post hoc test (Bonferroni’s test) showed an increase in the percentage of the distance traveled by the larvae in the central area in the control group (DMSO, *p* < 0.001), diazepam (*p* < 0.001) and isolaserpitin at 12.5 μM (*p* < 0.05), 25 μM (*p* < 0.001) and 50 μM (*p* < 0.05) in the dark phase, as compared with the light phase. In addition, during the dark phase of the experiment, significant increases in the distance traveled in the central area were observed when 5-dpf larvae were treated with diazepam (*p* < 0.001) and isolaserpitin at 12.5, 25 and 50 μM (*p* < 0.001), as compared with the DMSO-treated group in the dark phase (Figure 6b). During the light phase of the experiment, a significant increase in the percentage of the distance moved in the central arena was noticed when 5-dpf larvae were treated with diazepam (*p* < 0.001) and isolaserpitin at concentrations of 12.5 and 50 μM (*p* < 0.001), when compared to the DMSO-treated group in the light phase (Figure 6b).

In Figure 6c, the effect of different concentrations of isolaserpitin on thigmotaxis of the larvae is presented, when the duration of the activities in the central area was measured (two-way ANOVA light–dark condition [F(1, 320) = 201.56, *p* < 0.0001], interaction [F(4, 320) = 18.09, *p* < 0.0001] and treatment effect [F(4, 320) = 94.27, *p* < 0.0001]). The post hoc test (Bonferroni’s test) showed an increase in the percentage of the time spent by the larvae in the central arena in the control group (DMSO, *p* < 0.001), diazepam (*p* < 0.001) and isolaserpitin at concentrations of 12.5 μM (*p* < 0.05) and 25 μM (*p* < 0.001) in the dark phase, as compared with the light phase. In addition, during the dark phase of the experiment, significant increases in the time spent in the central arena were observed when 5-dpf larvae were treated with diazepam (*p* < 0.001) and isolaserpitin at 12.5, 25 and 50 μM (*p* < 0.001), as compared with the DMSO-treated group in the dark phase (Figure 6c). During the light phase of the experiment, a significant increase in the percentage of the time spent in the central arena was noticed when 5-dpf larvae were treated with diazepam (*p* < 0.001) and isolaserpitin at 12.5 and 50 μM (*p* < 0.001), when compared to DMSO-treated group in light phase (Figure 6c).

#### 2.2.6. The Effects of Ocatonoyllomatin on Locomotor Activity and Thigmotaxis Behaviors of the Zebrafish Larvae during Light-Dark Changes

Two-way ANOVA showed statistically significant changes in the light–dark condition response [F(1, 320) = 46.97, *p* < 0.0001], treatment effect [F(4, 320) = 4.62, *p* = 0.0016], as well as interaction [F(4, 320) = 4.97, *p* = 0.0009]. The post hoc Bonferroni’s test showed an increase in the locomotor activity during the dark phase in control group (DMSO, *p* < 0.01) and after incubation with octanoyllomatin at 9 and 15 μM (*p* < 0.001), in comparison with the light phase (Figure 7a). During the dark phase, significant decreases in the locomotor activity were observed when the larvae were treated with diazepam (*p* < 0.01) and octanoyllomatin at 30 μM (*p* < 0.01), as compared to the DMSO-treated group in the dark phase (Figure 7a).

The light–dark transitions were also studied for the thigmotaxis behavior on the zebrafish larvae. In Figure 7b it can be seen that the treatment with octanoyllomatin in the dark challenge influenced the thigmotaxis behaviors of the larvae, when the traveled distance was considered (two-way ANOVA light–dark condition [F(1, 334) = 312.3, *p* < 0.0001), treatment [F(7, 334) = 11.93, *p* < 0.0001] and interaction [F(7, 334) = 29.48, *p* < 0.0001]). The post hoc test (Bonferroni’s test) showed an increase in the percentage of the distance traveled by the larvae in the central area in control group DMSO (*p* < 0.05), diazepam (*p* < 0.01) and octanyollomatin at 9 μM (*p* < 0.01) and 15 and 30 μM (*p* < 0.001) in the dark phase, as compared with the light phase. In addition, during the dark phase of the experiment significant increases in the distance traveled in the central area were observed when 5-dpf larvae were treated with diazepam (*p* < 0.01) and octanyollomatin at 9 μM (*p* < 0.05), as compared with the DMSO-treated group in the dark phase (Figure 7b). During the light phase of experiment, a significant decrease in the percentage of the distance moved in the central arena was noticed when 5 dpf larvae were treated with diazepam (*p* < 0.05), as compared to the DMSO-treated group in the light phase (Figure 7b).

As mentioned before, except the distance traveled by the zebrafish larvae (5-dpf), the time spent in the central arena also plays a role in the indication of thigmotaxis and therefore of anxiolytic activity of tested compounds.

In Figure 7c, the effect of different concentrations of octanoyllomatin on thigmotaxis of the larvae is presented. The duration of the activities in the central area was measured (two-way ANOVA light–dark condition [F(1, 334) = 174.2, *p* < 0.0001], interaction [F(7, 334) = 20.71, *p* < 0.0001) and treatment effect [F(7, 334) = 6.847, *p* < 0.0001]). The post hoc test (Bonferroni’s test) showed an increase in the percentage of the time spent by larvae in the central arena in control group DMSO (*p* < 0.05), diazepam (*p* < 0.001) and octanyollomatin at 9, 15 and 30 μM (*p* < 0.001) in the dark phase, as compared with the light phase. In addition, during the dark phase of the experiment, significant increases in the time spent in the central arena were observed when 5-dpf larvae were treated with diazepam (*p* < 0.001) and octanyollomatin at 9 μM (*p* < 0.01) and 15 μM (*p* < 0.05), as compared with the DMSO-treated group in the dark phase (Figure 7c). During the light phase of the experiment, a significant increase in the percentage of the time spent in the central arena was noticed when 5-dpf larvae were treated with diazepam (*p* < 0.05) and decrease with octanyollomatin at 30 μM (*p* < 0.01), as compared to DMSO-treated group in light phase (Figure 7c).

## 3. Discussion

Apart from extraction, isolation is one of the most crucial phytochemical steps, as it strongly influences the outcomes of the subsequent bioactivity studies. Developing effective, fast and reproducible isolation methods is important not only for providing high-purity compounds, but also for ensuring sufficient amounts for further pharmacological assays. Support-free liquid–liquid chromatography, commonly referred to as countercurrent chromatography (CCC) and centrifugal partition chromatography (CPC), is a preparative separation technique in which the two phases of a biphasic solvent system are used as the mobile and stationary phase. One of the two phases is kept stationary inside the column with the help of a centrifugal field, while the other one is pumped through the column [21,22]. Since there is no solid sorbent, irreversible adsorption is practically eliminated. The costs of purification are relatively low, as expensive solid stationary phases, time intensive column packing procedures or high-purity solvents are not required [23,24]. Additionally, the ability to handle crude samples, ensure their maximum recovery and preserve their bioactivity makes liquid–liquid chromatography a versatile and highly adaptable preparative separation technique [18,24,25]. Both techniques have been proved their utility as an efficient method for isolation active coumarins from different plants from the Apiaceae family, including new structures [26,27,28]. The preliminary HPLC-DAD analysis of the SDMFE revealed the presence of several interesting coumarin compounds. The multi-solvent system HEMWat was previously shown to be efficient in separating compounds from other coumarin-rich sources [29,30,31,32]. Therefore, various HEMWat mixtures were tested in this work and, from these, HEMWat 2/5/2/5 (*v*/*v*/*v*/*v*) was selected for the separation of devenyol (**1**) and cis-khellactone (**2**), HEMWat 6/5/6/5 (*v*/*v*/*v*/*v*) for the separation of d-laserpitin (**3**) and isolaserpitin (**4**) and HEMWat 5/2/5/2 (*v*/*v*/*v*/*v*) for the separation of octanoyllomatin (**5**). Parallel batch liquid–liquid chromatography experiments with the three solvent systems were performed under identical operating conditions, allowing the one-step separation of high-purity coumarins. To supply the zebrafish assays with enough amounts of the high-purity coumarins, the liquid–liquid chromatography separations were repeated under the experimental conditions described in this section for as many times as needed.

Since all the five coumarins isolated in the current work by liquid–liquid chromatography were also purified by conventional solid–liquid chromatography (open column chromatography and preparative HPLC) in one/two steps from SDMFE fruits obtained under similar extraction conditions [14], we next roughly estimated the yields of the two techniques. It was noticed that the liquid–liquid chromatography yields varied from 1.6-fold to 17-fold higher as compared to the solid-liquid chromatography yields. We would like to point out that factors related to the variations in the chemical composition of the starting plant material (harvested in different years) could slightly shift the calculated yields of the two methods. Nevertheless, the biggest contribution for these differences would have to be due to the distinct separation mechanisms of solid–liquid chromatography and liquid–liquid chromatography.

Numerous experimental, as well as clinical data, demonstrate a direct link between the activation of the GABA-ergic neurotransmission and decreasing anxiety level and anxiety-like behaviors [6]. The general trend of seeking new substances for the pharmacotherapy of mental diseases (including anxiety) among medicinal plants has progressed significantly in the last two decades [33]. Numerous phytoconstituents, such as linalool, hypericin, chrysin, honokiol, quercitin, magnolol, apigenin, and many phenolic acids (*p*-coumaric acid, caffeic acid, ferulic acid, or sinapic acid), have shown promising anxiolytic activity [34]. The research has also not omitted plants containing coumarins as active compounds, especially these belonging to the Apiaceae family.

The anxiolytic properties of non-polar coumarins imperatorin and isoimperatorin were established in an animal model using the elevated plus maze (EPM). The tested substances were found to have significant anxiolytic activity expressed by an increase in the percentage of time and number of arms entries in the open arm [35,36]. Previous studies showed that prenyl residues attached to the basic coumarin skeleton are essential for the positive modulatory activity, while longer side chains or big residues (e.g., geranyl residues) significantly reduce GABA modulation [37]. Among tested coumarins, phellopterin (a furanocoumarin with a lipophilic methoxy group at C5 and 3-methyl-2-butyneloxy side chain attached at C8) appears to be one of the most potent compounds with very high selectivity (IC_50_ = 0.36 μM and reduction of side effects) [9,38]. Furanocoumarins deprived of these substituents are characterized by weaker influence on GABA_A_ receptor, e.g., imperatorin which lacks the methoxy group at C5 or bergapten which does not have a side chain attached at C8 [9]. Interestingly, some other furanocoumarins, such as phellopterin, psolaren, byakangelicol, indicated to act as partial agonists at GABA/BDZ receptor [39].

Previously, we reported that the mature fruits of *Seseli devenyense* are an abundant source of coumarin compounds with an interesting structure [14]. Therefore, we decided to evaluate the effect of its methanolic extract (SDMFE) on the on the anxiety behavior (anxiolytic activity) in a zebrafish larvae model. The vast majority of studies on the anxiolytic action of natural or synthetic substances are carried out using rodents [40]. However, the number of experiments based on the use of larvae is increasing. The use of the *Danio rerio* animal model to study anxiolytic activity has great advantages. Small size, rapid development, and cost-effective assays allow in vivo high-throughput screenings of compounds that modulate behavioral responses [41]. In this context, the zebrafish model was previously used for the evaluation of the anxiolytic activity of xanthotoxin, a furanocoumarin isolated from the fruits of *Angelica archangelica* [42]. In this comparative study, the utility of the zebrafish model was collated with the well-known murine model, in which thigmotaxis, the tendency to remain close to the walls, was used as an index of anxiety. It has been previously shown that thigmotactic behaviors (anxiety-like behaviors) can be attenuated and enhanced by commonly used anxiolytics (BDZ, e.g., diazepam) or anxiogenic substances (caffeine), respectively, in both adult and larval zebrafish [43]. The results obtained by Maciag et al. (2020) showed the same effects of xanthotoxin in the two different animal models and confirmed the strong activity of xanthotoxin, which exerted reversed U-shape effect on anxiety behaviors. Therefore, these findings highlighted the predictive power of the zebrafish model for evaluating the anxiolytic activity of natural compounds [42].

Consequently, the zebrafish model was used in our work for the evaluation of the anxiolytic activity of the SDMFE and its isolated coumarins. Our findings show that the acute administration of SDMFE (3–12.5 μg/mL) generally increased the locomotor activity of zebrafish larvae during 40 min period of continuous light. The changing ratios between the inner (central) and outer arena, in favor for the central arena, as evaluated by the time spent and distance moved in the central arena, indicated a decreasing thigmotaxis level and an increase in the anxiolytic activity, especially at higher concentrations.

The zebrafish larvae abruptly exposed to transition from light to darkness strongly increased the locomotor activity and enhanced the exploration and escape behavior [43]. As a part of the repertoire of anxiety-like behaviors, thigmotaxis increased and this transition can be regarded as an anxiety trigger [41]. Furthermore, SDMFE (9 and 25 μg/mL) increased the distance moved in the central arena, as compared to the control group (DMSO-treated) during the dark phase of the experiment, which indicated the anxiolytic activity. When considering the time spent in the central arena, there was no significant increase in the percentage of the time spent in central arena, in comparison to the control group in the dark phase.

Since SDMFE showed a very promising anxiolytic activity in the zebrafish larvae model, the extract was used as raw material for the isolation of high-purity coumarin compounds, namely devenyol (simple hydroxycoumarin) and cis-khellactone, d-laserpitin, isolaserpitin and octanyllomatin (pyranocoumarins).

Besides isolaserpitin, all remaining coumarins had no significant influence on the spontaneous locomotor activity and thigmotaxis behavior (both distance moved and time spent in the central area) during 40 min of continuous illumination at all concentrations (Appendix A). However, the anxiolytic effects were observed under stress conditions (during light on/off phases). In the second part of the experiments (during the light–dark challenge), diazepam as well as all devenyol, d-laserpitin, isolaserpitin and octanoyllomatin significantly increased the distance moved by the larvae in the central arena and the time spent in the central area, in the dark phase of the experiment as compared to the light phase.

According to our knowledge, these results show for the first time the anxiolytic activity of simple coumarins isolated from the Apiaceae family in a zebrafish model of anxiety. Literature data mentioned a few simple coumarins showing anxiety-related activity. For example, ostruthin [6-(3,7-dimethyl-2,6-octadienyl)-7-hydroxycoumarin], consistently showed an anxiolytic effect on mice at 5 mg/kg. The mechanism of action was associated with activation of the TREK-1 potassium channel and suppression of anxiety-related neurons [44]. Bio-guided fractionation from the root extract of *Biberstenia multifida* DC (Geraniaceae) indicated significant sustained anxiolytic effects of simple hydroxycoumarins, such as umbelliferone (7-hydroxycoumarin) and scopoletin (7-hydroxy-6-methoxycoumarin) [45]. Moreover, daphnoretin, a bicoumarin of scopoletin and umbelliferone, exhibited a significant anxiolytic activity in a rodent model of anxiety (tested on EPM) [46]. It could be noticed that the free hydroxyl group at the C7 of the benzopyron skeleton might be crucial for the anxiolytic potential. This hypothesis is support by our results which showed that devenyol, a C7-hydroxyderivative, was active at concentrations of 1.5, 6, 9 and 15 μM.

Based on the data obtained in our study, several structure-activity relations were valiantly proposed for the four pyranocoumarins isolated from *Seseli devenyense*. The anxiolytic effect of cis-khellactone derivatives might be influenced by the esterification at C3’ or C4’ of the pyran ring. Laserpitin and isolaserpitin are both esters of cis-khellactone with angelic acid (2Z)-2-methylbut-2-enoic acid), one at C4’ and the other at C3’, respectively; however, it was noticed that the anxiolytic activity was higher when the esterification occurred at C3’ (isolaserpitin). Additionally, octanoyllomatin, a dehydroxy-cis-khellactone derivative esterified at C3’ with caprylic acid, showed the highest anxiolytic activity. The anxiolytic activity of this lomatin derivative also correlates with the highest lipophilic character provided by the long aliphatic chain and lack of a free hydroxy group at C4’. Thus, our data showed that both the position and nature of the esther group attached at the pyran ring have a significant contribution to the observed behavior of the zebrafish larvae. Lastly, cis-khellactone, the non-esterified coumarin, exhibited the weakest activity.

To the best of our knowledge, this is the first report where series of pyranocoumarin derivatives were shown to be endowed with promising anxiolytic activity. A thorough search of the relevant literature revealed that the traditional use *Saposhnicovia divaricata* (Turcz.) Schischk (Apiaceae) in the treatment of anxiety occurring in children might be linked to the presence of pyranocoumarins [47].

## 4. Materials and Methods

### 4.1. Chemicals

Analytical-grade hexane, ethyl acetate and methanol (POCh, Gliwice, Poland) were used for the extraction and isolation studies. Chromatographic grade methanol, acetonitrile, water, ammonium formate and formic acid (J.T. Baker, Deventer, The Netherlands) were used for the HPLC-DAD and HPLC-ESI-QTOF-MS/MS analyses. Deuterated chloroform (CDCl_3_, 99.8 atom %D) for NMR analyses was acquired from Armar Chemicals (Döttingen, Switzerland). Water was purified using a Simplicity^®^ water purification system (Millipore, Grenoble, France). Dimethyl sulfoxide (DMSO) for molecular biology (Sigma-Aldrich, Darmstadt, Germany). E3 solution for *Danio rerio* experiments (solution of a mixture of salt: NaCl, KCl, CaCl_2_, MgCl_2_).

### 4.2. Plant Material and Extraction

The mature fruits of *Seseli devenyense* Simonk. were harvested from the Botanical Garden of Maria Curie-Sklodowska University (UMCS, Lublin, Poland). The plant material was identified by dr Agnieszka Dąbrowska (UMCS, Lublin, Poland). A voucher specimen (No. 49_2017) was deposited in the Department of Pharmacognosy, Medical University of Lublin (Lublin, Poland). The collected fruits were separated from umbels, air-dried, grounded to powder and sieved on a 1.0 mm meshes sifter. Then, the plant material (30 g) was extracted with methanol in a Soxhlet apparatus, under the reflux for 48 h, affording 1.5 g of dried crude extract (yield: 5.0%).

### 4.3. Liquid–Liquid Chromatography

#### 4.3.1. Shake-Flask Experiments

The screening for suitable biphasic solvent systems to isolate the target coumarins from *Seseli devenyense* methanolic fruit extract (SDMFE) was performed by shake-flask experiments. The partition coefficient (*Pi*), defined as the ratio of the concentration of a solute in the stationary phase to its concentration in the mobile phase, was used as the screening parameter. The experimental protocol was previously descried by Widelski et al. (2018) [30]. Briefly, 1 mg of crude extract was dissolved in 4 mL of biphasic solvent systems obtained by mixing various volumes of hexane, ethyl acetate, methanol and water (HEMWat). After vigorous shaking, the flasks were left until the complete separation of the layers. Then, equal volumes of the upper and lower phase (0.5 mL) were taken, evaporated to dryness, dissolved again in methanol (1.0 mL) and analyzed by HPLC-DAD analysis, as descried below.

#### 4.3.2. Preparation of Biphasic Solvent Systems

The chosen biphasic solvent systems were prepared in separation funnels by mixing adequate volumes of each solvent. The mixtures were then vigorously shaken and equilibrated at room temperature. Then, the upper and lower phases were split 15 min before use, labelled in different containers and degassed in ultrasound baths for 10 min.

#### 4.3.3. Liquid–Liquid Chromatography Separations

The liquid–liquid chromatography experiments were performed on a countercurrent chromatograph (CCC) from Dynamic Extractions (Slough, UK) equipped with polytetrafluoroethylene multilayer coils for both analytical (0.8 mm i.d., 22 mL column volume) and semi-preparative (1.6 mm i.d., 136 mL column volume) scale experiments. The CCC was connected to a Sapphire UV detector and Alpha 10 pump (ECOM, Prague, Czech Republic). All separations were carried out in reversed-phase mode (head-to-tail, descending mode). At the beginning of each experiment, the semi-preparative column was filled with stationary upper phase. Then, the rotational speed was set at 1600 rpm and the mobile lower phase was pumped through the column at 6 mL/min, until no more stationary phase eluted from the column. Then, samples (40 mg/mL dissolved in equal volumes of upper and lower phases) were injected via a 6 mL loop. All separations were manually fractionated (1 min) at 320 nm. The collected fractions were analyzed by HPLC-DAD; fractions containing high-purity coumarins were pooled together and evaporated to dryness.

### 4.4. HPLC-DAD Analysis

The preliminary analysis of the SDMFE, determination of the partition coefficient values and purity of collected fractions were assessed by HPLC-DAD. The HPLC-DAD analyses were performed on a Shimadzu HPLC (Shimadzu, Tokyo, Japan) equipped with automatic degasser (DGU-20A3R), quaternary pump (LC-20AD), auto-sampler (SIL-20AHT) and diode array detector (SPD-M20A). The separations were carried out on an Agilent Zorbax Eclipse XDB-C18 (Agilent, Santa Clara, CA, USA) (250 mm × 4.6 mm, 5 μm) column with a mobile phase consisting of water (solvent A) and methanol (solvent B). The following gradient was applied: 0 min: 50% B; 5 min: 60% B; 25 min: 80% B; 30–35 min: 100% B. The flow-rate was 1 mL/min; the column temperature was set at 25 °C. The injection volume of sample was 10 μL and detection was carried out at 320 nm.

### 4.5. Structural Elucidation

The full structural elucidation of isolated compounds was performed by high-performance liquid chromatography hyphenated with electrospray ionization-quadrupole-time-of-flight-tandem mass spectrometry (HPLC-ESI-QTOF-MS/MS) and nuclear magnetic resonance (NMR). HPLC-ESI-QTOF-MS/MS analysis was performed on an Agilent 1200 HPLC (Agilent Technologies, Santa Clara, CA, USA) equipped with degasser (G1379B), binary pump (G1312C), thermostated column compartment (G1316A), auto-sampler (G1329B), diode array detector (G1315D) and ESI-QTOF mass spectrometer (G6530B). The separations were carried out on a Zorbax Stable Bond RP-18 (250 mm × 2.1 mm, 5 μm) column with a mobile phase consisting of 0.1% formic acid + 10 mM ammonium formate (pH = 3.5) + 1% acetonitrile in water (solvent A) and 0.1% formic acid + 10 mM ammonium formate (pH = 3.5) + 95% acetonitrile in water (solvent B). The following gradient was applied: 0–10 min: 40% B; 30–35 min: 80% B; 40–45 min: 100% B. The following MS parameters were used: positive ionization mode; mass range: 100–1000 *m*/*z*; gas temperature 350 °C; nitrogen flow: 10 L/min; nebulizer pressure 40 psi; skimmer 65 V; capillary voltage 3500 V; fragmentor 100 V; collision-induced dissociation energy 10 V. For NMR analyses, a Brucker Avance III HD 600 MHz spectrometer with QCI 5 mm Cryoprobe and SampleJet automated sample changer (Bruker BioSpin, Rheinstetten, Germany) was used. The structural characterization was achieved in CDCl_3_ by 1D-NMR (^1^H-NMR; ^13^C-DEPTQ-NMR) and 2D-NMR (correlation spectroscopy, COSY; heteronuclear multiple-bond correlation, HMBC; multiplicity-edited heteronuclear single-quantum correlation, edited-HSQC; rotating-frame Overhauser enhancement spectroscopy, ROESY).

### 4.6. Zebrafish Model for Evaluation Anxiety-Like Behaviors

#### 4.6.1. Zebrafish Husbandry and Egg Collection

*Danio rerio* stocks of the wild type zebrafish strain (AB strain, Experimental Medicine Centre, Medical University of Lublin, Poland) were maintained at temperature 26–28.5 °C in a controlled environment (pH ranging between 6.9 and 7.5; conductivity of 550–700; 14/10 h light/dark cycle). Embryos were reared under standard light/day cycle in E3 embryo medium (pH 7.1–7.3; 17.4 μM NaCl, 0.21 μM KCl, 0.12 μM MgSO4 and 0.18 μM Ca(NO_3_)_2_) in an incubator (IN 110 Memmert GmbH, Buechenbach, Germany). The 5-dpf zebrafish larvae were used for the assays. After the experiment, larvae were immediately killed by immersion in a solution of tricaine (15 μM). All experiments were conducted in accordance with the National Institute of Health Guidelines for the Care and Use of Laboratory Animals and to the European Community Council Directive for the Care and Use of Laboratory Animals of 22 September 2010 (2010/63/EU). For the experiment with larvae up to 5-dpf, agreement of Local Ethical Commission is not required.

#### 4.6.2. Determination of the Maximum Tolerated Concentration

Stocks solution of 20 mg/mL of crude extracts and 50 mM of pure compounds as well as 100 mM for diazepam were prepared in DMSO, dissolved (by sonication) and stored in the fridge at −20 °C.

Firstly, maximum tolerated concentration (MTC) for each compound and the extract was determined. For this purpose, 5-dpf zebrafish larvae were incubated for 18 h in dark environment at 28 °C with the tested compounds (0.5 mL of solution to each well) in 4 concentrations—100 μM, 50 μM, 25 μM and 12.5 μM for pure compounds and 100 μg/mL, 50 μg/mL, 25 μg/mL and 12.5 μg/mL for the crude extract (SDMFE), respectively, in 48 well-plates. Five larvae were placed in each of the well, which gives a total of 10 larvae per concentration. As a control, 1% DMSO in E3 solution was used. The following symptoms of acute locomotor impairment were evaluated: hypoactivity, no touch response, decreased touch response, loss of posture, body deformation, exophthalmos, slow heartbeat, oedema, and finally dead. MTC was determined based on those results.

#### 4.6.3. Anxiolytic Activity Assay

The research protocol was adopted after Schnorr et al. (2012). The anxiolytic activity was evaluated in 24-well plates. The inner and outer zones of the well were delineated according to Schnorr et al. (2012) [41]. The anxiolytic effect was defined as an increase of the time spent in the inner zone in comparison with the control group. It could be also described as decreasing thigmotaxis. 1% DMSO in E3 solution was used as the negative control, 10 μM diazepam was used as the positive control. 32 larvae were used for each concentration.

The experimental procedure was divided into three stages:(a)Preincubation: larvae were pre-incubated in a 1500 μL solution of tested compounds in individual wells of a 24-well plate at 28.5 °C, in dark, for 30 min.(b)Acclimatization (min 0–10): plates were placed in an automated video recording bench station (ViewPoint, Lyon, France). Larvae were allowed to acclimatize to the test apparatus for 10 min. During this period, lights were kept on.(c)Visual motor challenge (min 11–95): this stage experiment was divided into two phases:Phase 1—continuous illumination: larvae were subjected to continuous light for 40 min, with the purpose to determine the spontaneous locomotor activity and thigmotaxis behavior (before applying the factor causing the anxiety).Phase 2—light/dark transition: larvae were subjected to three cycles, each involving the 10-min lighting phase (continuous light) and the 5-min dark phase (anxiogenic factor).

### 4.7. Statistical Analysis

For the determination of the statistical differences between various tested substances and their concentrations, the Prism software (GraphPad Software, San Diego, CA, USA) was used. Data were presented as mean ± standard error of the mean (SEM). For comparison, the pretreatment and treatment effects obtained data were analyzed using analysis of variance (one-way or two-way ANOVA). One-way ANOVA was followed by the Tukey’s test (post hoc test). In the case of two-way ANOVA, Bonferroni’s test was used as post hoc test. The confidence limit of *p* < 0.05 was considered as statistically significant.

## 5. Conclusions

The abovementioned research had two main goals. The first task was to isolate coumarin compounds from the mature fruits of *Seseli devenyense* Simonk. (Apiaceae). For that work, a modern separation technique, namely, liquid–liquid chromatography, was used, leading to high-purity simple coumarins (devenyol) and pyranocoumarins (cis-khellactone, d-laserpitin, isolaserpitin, octanoyllomatin). The evaluation of the anxiolytic activity of isolated coumarins on zebrafish larvae model was the second task of this work. Therein, we reported for the first time the anxiolytic activity of four compounds belonging to pyranocoumarins. Furthermore, our results confirmed the usefulness of the zebrafish model to evaluate the influences of natural substances on anxiety behaviors. Overall, the efficiency and speed of isolation using liquid–liquid chromatography was thoroughly combined with the utility and reproducibility of the animal model (*Danio rerio*). The resulting test platform is suitable for large-scale screening studies.

## Figures and Tables

**Figure 1 ijms-22-01829-f001:**
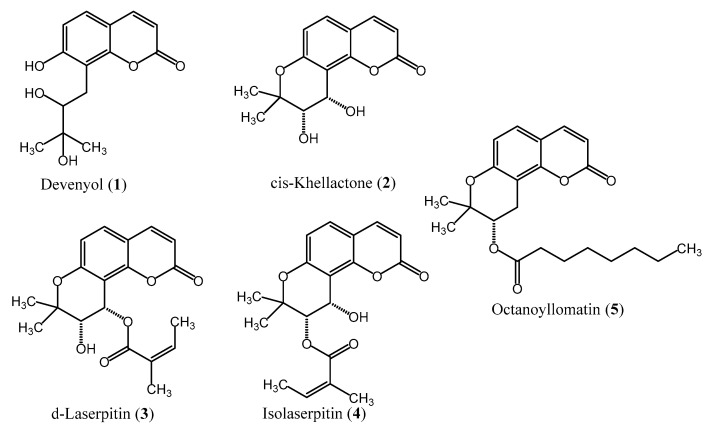
Chemical structures of isolated coumarins.

**Figure 2 ijms-22-01829-f002:**
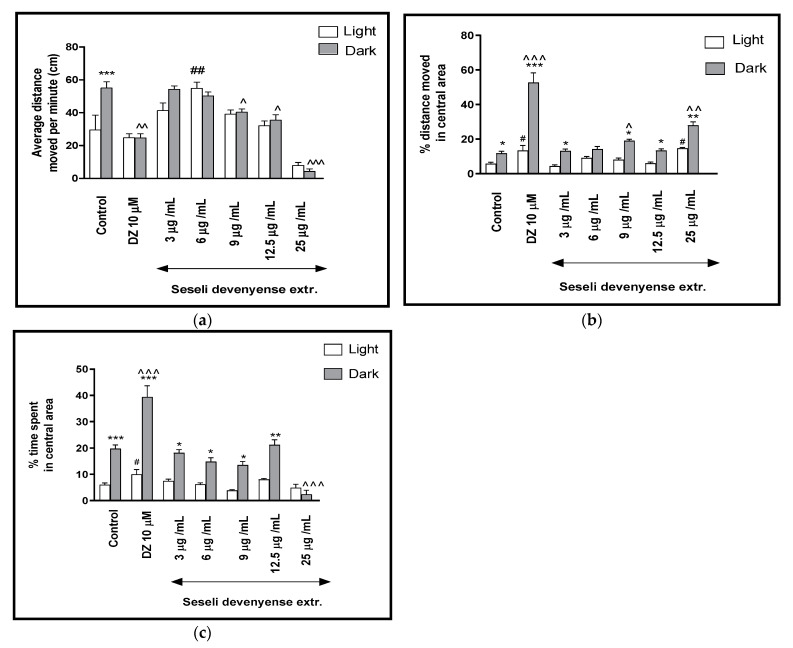
The effects of SDMFE (3, 6, 9, 12.5, 25 μg/mL) and diazepam (DZ, 10 μM) on locomotor activity during all three light–dark challenge phase. (**a**) Average distance moved by zebrafish larvae within each 1-min time bin under either light (open bars) or dark (filled bars); (**b**) The percentage of distance moved by zebrafish larvae in the central area during light phase (white bars) or dark phase (filled bars); (**c**) The percentage of time spent by zebrafish larvae in the central area during light phase (white bars) or dark phase (filled bars). Data are presented as mean ± SEM; n = 32. * *p* < 0.05, ** *p* < 0.01, *** *p* < 0.001 in comparison to light conditions within the same concentration group; ^ *p* < 0.05, ^^ *p* < 0.01, ^^^ *p* < 0.001 in comparison to control group under dark condition; # *p* < 0.05, ## *p* < 0.01 in comparison to control group under light condition (post hoc Bonferroni’s test).

**Figure 3 ijms-22-01829-f003:**
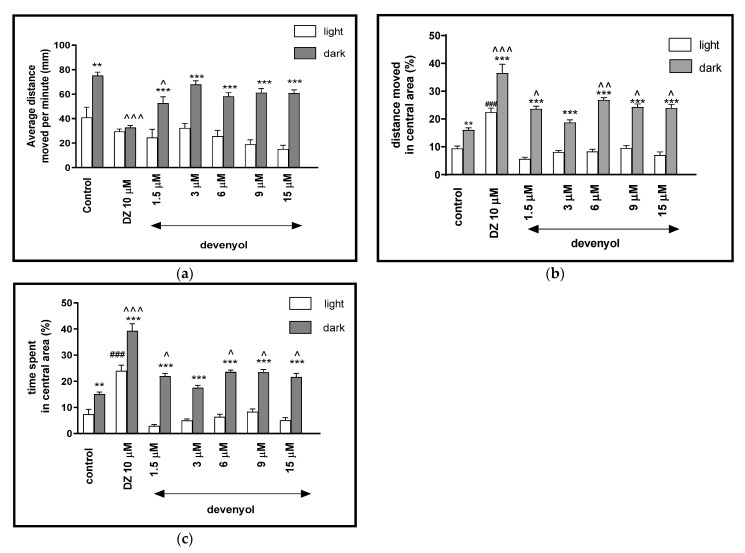
The effects of devenyol (1.5, 3, 6, 9 and 15 μM) and diazepam (DZ, 10 μM) on locomotor activity during all three light–dark challenge phase. (**a**) Average distance moved by zebrafish larvae within each 1-min time bin under either light (open bars) or dark (filled bars). (**b**) The percentage of distance moved by zebrafish larvae in the central area during light phase (white bars) or dark phase (filled bars). (**c**) The percentage of time spent by zebrafish larvae in the central area during light phase (white bars) or dark phase (filled bars). Data are presented as mean ± SEM; n = 32. ** *p* < 0.01, *** *p* < 0.001 in comparison to light conditions within the same concentration group; ^ *p* < 0.05, ^^ *p* < 0.01, ^^^ *p* < 0.001 in comparison to control group under dark condition; ### *p* < 0.001 in comparison to control group under light condition (post hoc Bonferroni’s test).

**Figure 4 ijms-22-01829-f004:**
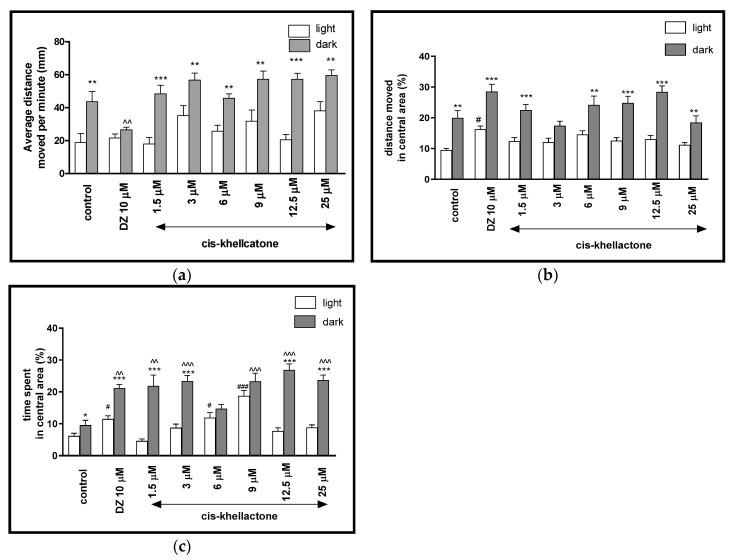
The effects of cis-khellactone (1.5, 3, 6, 9, 12.5 and 25 μM) and diazepam (DZ, 10 μM) on locomotor activity during all three light–dark challenge phase. (**a**) Average distance moved by zebrafish larvae within each 1-min time bin under either light (open bars) or dark (filled bars). (**b**) The percentage of distance moved by zebrafish larvae in the central area during light phase (white bars) or dark phase (filled bars). (**c**) The percentage of time spent by zebrafish larvae in the central area during light phase (white bars) or dark phase (filled bars). Data are presented as mean ± SEM; n = 32. * *p* < 0.05, ** *p* < 0.01, *** *p* < 0.001 in comparison to light conditions within the same concentration group; ^^ *p* < 0.01, ^^^ *p* < 0.001 in comparison to control group under dark condition; # *p* < 0.05, ### *p* < 0.001 in comparison to control group under light condition (post hoc Bonferroni’s test).

**Figure 5 ijms-22-01829-f005:**
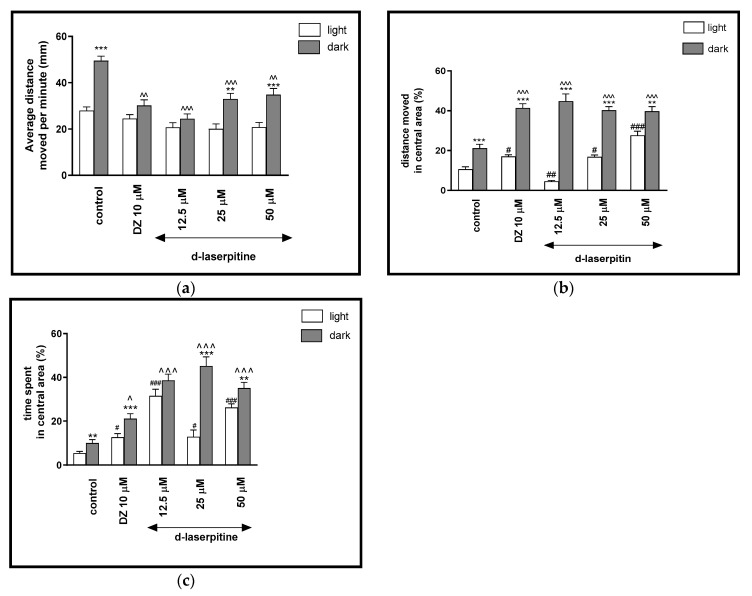
The effects of d-laserpitin (12.5, 25 and 50 μM) and diazepam (DZ, 10 μM) on locomotor activity during all three light–dark challenge phase. (**a**) Average distance moved by zebrafish larvae within each 1-min time bin under either light (open bars) or dark (filled bars). (**b**) The percentage of distance moved by zebrafish larvae in the central area during light phase (white bars) or dark phase (filled bars). (**c**) The percentage of time spent by zebrafish larvae in the central area during light phase (white bars) or dark phase (filled bars). Data are presented as mean ± SEM; n = 32. ** *p* < 0.01, *** *p* < 0.001 in comparison to light conditions within the same concentration group; ^ *p* < 0.05, ^^ *p* < 0.01, ^^^ *p* < 0.001 in comparison to control group under dark condition; # *p* < 0.05, ## *p* < 0.01, ### *p* < 0.001 in comparison to control group under light condition (post hoc Bonferroni’s test).

**Figure 6 ijms-22-01829-f006:**
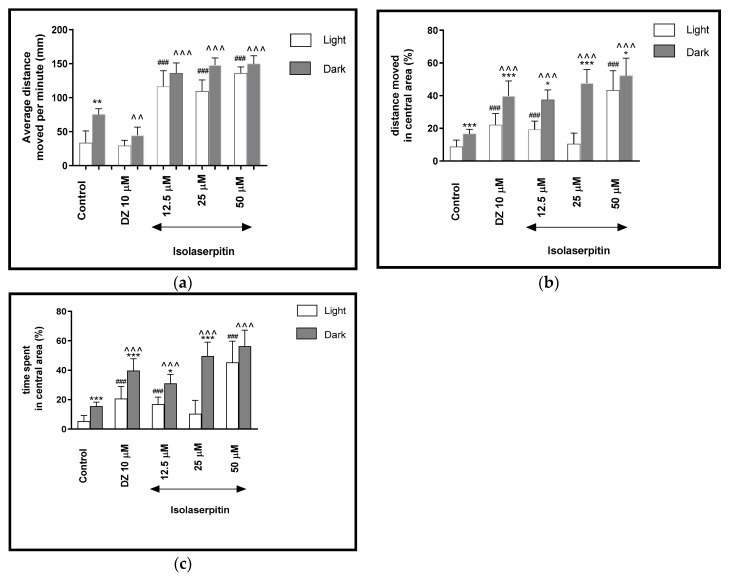
The effects of isolaserpitin (12.5, 25 and 50 μM) and diazepam (DZ, 10 μM) on locomotor activity during all three light–dark challenge phase. (**a**) Average distance moved by zebrafish larvae within each 1-min time bin under either light (open bars) or dark (filled bars). (**b**) The percentage of distance moved by zebrafish larvae in the central area during light phase (white bars) or dark phase (filled bars). (**c**) The percentage of time spent by zebrafish larvae in the central area during light phase (white bars) or dark phase (filled bars). Data are presented as mean ± SEM; n = 32. * *p* < 0.05, ** *p* < 0.01, *** *p* < 0.001 in comparison to light conditions within the same concentration group; ^^ *p* < 0.01, ^^^ *p* < 0.001 in comparison to control group under dark condition; ### *p* < 0.001 in comparison to control group under light condition (post hoc Bonferroni’s test).

**Figure 7 ijms-22-01829-f007:**
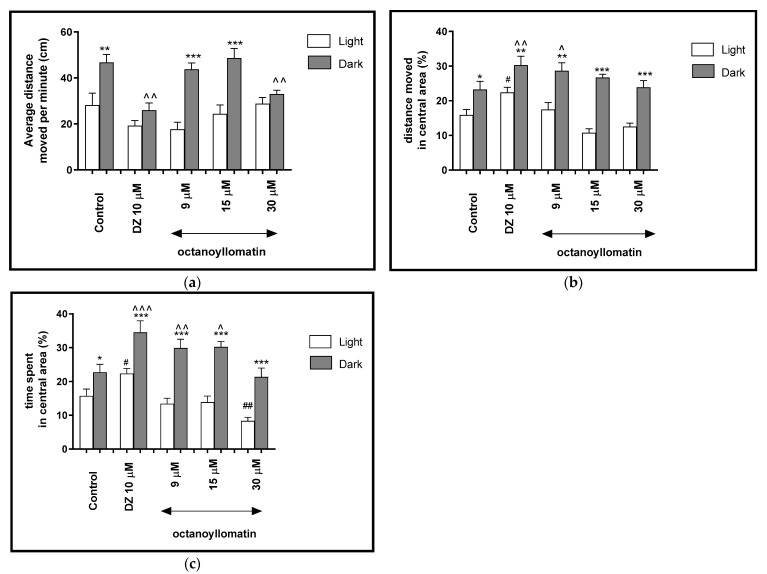
The effects of octanoyllomatin (9, 15 and 30 μM) and diazepam (DZ, 10 μM) on locomotor activity during all three light–dark challenge phase. (**a**) Average distance moved by zebrafish larvae within each 1-min time bin under either light (open bars) or dark (filled bars). (**b**) The percentage of distance moved by zebrafish larvae in the central area during light phase (white bars) or dark phase (filled bars). (**c**) The percentage of time spent by zebrafish larvae in the central area during light phase (white bars) or dark phase (filled bars). Data are presented as mean ± SEM; n = 32. * *p* < 0.05, ** *p* < 0.01, *** *p* < 0.001 in comparison to light conditions within the same concentration group; ^ *p* < 0.05, ^^ *p* < 0.01, ^^^ *p* < 0.001 in comparison to control group under dark condition; # *p* < 0.05, ## *p* < 0.01 in comparison to control group under light condition (post hoc Bonferroni’s test).

**Table 1 ijms-22-01829-t001:** Partition coefficient values of the target coumarins from *Seseli devenyense* methanolic fruit extract.

Hexane/Ethyl Acetate/Methanol/Water (*v*/*v*/*v*/*v*)		Partition Coefficients (*P_i_*)				
		*P* _1_	*P* _2_	*P* _3_	*P* _4_	*P* _5_
I	2/5/2/5	0.80	2.50	–	–	–
II	6/5/6/5	–	–	0.84	1.01	
III	5/2/5/2	–	–	–	–	2.01

**Table 2 ijms-22-01829-t002:** Comparison of the outcomes obtained from the liquid–liquid chromatography and solid-liquid chromatography separations of individual coumarins from *Seseli devenyense* methanolic fruit extract.

Technique	Mass of Processed Sample	Mass of Isolated Coumarin (Yield ^§^)					Total Separation Time ^$^	Total Solvent Consumption ^#^
		1	2	3	4	5		
Liquid–liquid chromatography	240 mg	5.5 mg (2.292%)	9.2 mg (3.833%)	1.4 mg (0.583%)	1.4 mg (0.583%)	3.1 mg (1.292%)	~3.5 h	~1.3 L
Solid–liquid chromatography *	1.6 g	7 mg (0.437%)	3.6 mg (0.225%)	2 mg (0.125%)	3 mg (0.187%)	13 mg (0.812%)	ns	ns

* data for the solid–liquid chromatography separations were extracted from [14,19,20]; ^§^ the yields were calculated by dividing the mass of each individual coumarin obtained to the mass of sample (extract) processed; ^$^ the total separation time was calculated by summing up the duration of the three liquid–liquid chromatography separations; ^#^ the total solvent consumption was calculated by multiplying the total separation time with the flow-rate; ns, not specified.

## Data Availability

The data presented in this study are available on request from the corresponding author.

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
