# Peer review of "Coumarins from Seseli devenyense Simonk.: Isolation by Liquid–Liquid Chromatography and Potential Anxiolytic Activity Using an In Vivo Zebrafish Larvae Model"

_ijms, 2021, doi:10.3390/ijms22041829_

Round 1

Reviewer 1 Report

The authors describe an article entitled "Coumarins from Seseli devenyense Simonk.: Isolation by liquid-liquid chromatography and potential anxiolytic activity using an in vivo zebrafish larvae model”. The topic of the manuscript is interesting, and the manuscript constitutes an interesting article concerning the isolation of natural products.

The work is well written, and sufficient spectra and figures are included in the manuscript for comprehension and clarity. Overall, I think that this is a good manuscript, that I recommend for publication after inclusion of minor revisions.

1) The authors mentioned that from 30 g of plant materials, 1.5 g of dried crude extract could be obtained. Using this procedure, all the coumarins contained in the plant materials were extracted or not ?

2) What about the photochemical stability of the coumarins ? This point should be commented.

3) Could the authors mention in which natural plants, the different coumarins can be found ? is the amount of each coumarin higher than that found in mature fruits of Seseli devenyense Simonk ?

For the abovementioned reasons, at present do not publish.

Author Response

1) The authors mentioned that from 30 g of plant materials, 1.5 g of dried crude extract could be obtained. Using this procedure, all the coumarins contained in the plant materials were extracted or not ?

2) What about the photochemical stability of the coumarins ? This point should be commented.

3) Could the authors mention in which natural plants, the different coumarins can be found ? is the amount of each coumarin higher than that found in mature fruits of Seseli devenyense Simonk ?

Response 1: Please provide your response for Point 1. (in red)

Extraction plant material in Soxhlet apparatus with methanol let to obtain , in our opinion all coumarins presented in plant material. The chromatographic profile (HPLC-DAD) of the crude extract indicated the dominant compounds, which were isolated using the counter-current chromatography technique.

Response 2: Photochemical stability in the case of coumarin compounds is of paramount importance.

In order to avoid the effects of light radiation, the isolated compounds were stored in a dark place.

In addition, HPLC-DAD analysis was performed several times during the experiment, which confirmed not only the purity of the compounds but also the absence of degradation products that could occur under the influence of light.

Response 3: Devenyol (simple coumarin) was isolated from fuits of Seseli devenyense for the first time (new natural compund), so we can not idicate another sourse of this particular coumarin. The rest of isolated and tested compunds occur in numerous representatives of Apiaceae family , esspecially in genera Seseli, Peucedanum, Angelica and others. For example, d-Laserpitin was isolated among others pyranocumarins from aerial part  of  Seseli gummiferum subsp.corymbosum. From 1 kg of plant material after extraction in Soxhlet apparatus 85 g of hexane extract was obtained. From this amount of extract 77.8 mg o d-Laserpitine was isolated (yield about 0.091 %). Our results concerning soli-liquid  column chromatography on silica gel are close (0,125 %), but liquid-liquid (counter-current) chromatography gives more than 6-fold higher results (0,583 % of yield for d-Laserpitin).  From lyophilized and powdered exudate (4.93 g) obtained from the stems of Angelica keiskei after MeOH–H2O soluble fraction (4.55 g) was obtained. 1.4 mg of D-Laserpitin and 31.2 mg of isolaserpitin were isolated from this fraction on preparative HPLC. The yield of both coumarin, esspecially isolaserpitin is better.

Reviewer 2 Report

The manuscript with the tittle Coumarins from Seseli devenyense Simonk.: Isolation by liquid-liquid chromatography and potential anxiolytic activity using an in vivo zebrafish larvae model. The topic is interesting and worthy of publication. Some corrections and explanations (listed below) are needed:

  1. Why the dot is written after Simonk. In the tittle as well as in the text?
  2. Page 2; line 97 you have title Result and discussion and on page 18; line 471 you have tittle Discussion. Corrected, the results should be presented in the results section and the discussion should be in the discussion section.
  3. Page 3; line 99-100; The preliminary HPLC-DAD analysis of the Seseli devenyense methanolic fruit extract 99 (SDMFE) revealed the presence of several interesting coumarin compounds, this sentence need to be re-write, chromatogram displays a "peak" and each peak in the chromatogram indicates the presence of a compound in the sample.
  4. Figure 1 and Figure 2 can be reduced.
  5. Page 5; line 142; “The spectroscopic data (Supplementary material – Part B) were in agreement with those available in spectral libraries and literature (14)”. Write literature source of spectral libraries and literature. Following coumarin derivatives are published in literature source 14: (+)-Hexanoyllomatin, (+)-Octanoyllomatin, (+)-Decanoyllomatin, (+)-Dodecanoyllomatin, (+)-4′-Decanoyl-cis-khellactone, (+)-3′-Decanoyl-cis-khellactone, (-)-8-[(2S),3-Dihydroxy-3-methylbutyl]-7-hydroxychromen-2-one, 2′-epi-Tortuoside: (-)-3′-O-b-D-glucopyranoside of 8-[(2S),3-dihydroxy-3-methylbutyl]-7-hydroxychromen-2-one
  6. Page 5; line 146; “from Widelski et al. (2005)” correct to Widelski et al. (14).
  7. Page 5; Table 2; d-Laserpitin and Isolaserpitin are not mention in reference 14.
  8. In Figure 4-Figure 9, m is bold corrected

Author Response

  1. Why the dot is written after Simonk. In the tittle as well as in the text?

  1. Page 2; line 97 you have title Result and discussion and on page 18; line 471 you have tittle Discussion. Corrected, the results should be presented in the results section and the discussion should be in the discussion section.
  2. Page 3; line 99-100; The preliminary HPLC-DAD analysis of the Seseli devenyense methanolic fruit extract 99 (SDMFE) revealed the presence of several interesting coumarin compounds, this sentence need to be re-write, chromatogram displays a "peak" and each peak in the chromatogram indicates the presence of a compound in the sample.
  3. Figure 1 and Figure 2 can be reduced.
  4. Page 5; line 142; “The spectroscopic data (Supplementary material – Part B) were in agreement with those available in spectral libraries and literature (14)”. Write literature source of spectral libraries and literature. Following coumarin derivatives are published in literature source 14: (+)-Hexanoyllomatin, (+)-Octanoyllomatin, (+)-Decanoyllomatin, (+)-Dodecanoyllomatin, (+)-4′-Decanoyl-cis-khellactone, (+)-3′-Decanoyl-cis-khellactone, (-)-8-[(2S),3-Dihydroxy-3-methylbutyl]-7-hydroxychromen-2-one, 2′-epi-Tortuoside: (-)-3′-O-b-D-glucopyranoside of 8-[(2S),3-dihydroxy-3-methylbutyl]-7-hydroxychromen-2-one
  5. Page 5; line 146; “from Widelski et al. (2005)” correct to Widelski et al. (14).
  6. Page 5; Table 2; d-Laserpitin and Isolaserpitin are not mention in reference 14.
  7. In Figure 4-Figure 9, m is bold corrected

Response 1: Seseli devenyense Simonk. is full botanical name (including dot after Simonk), according with the present systematic and official sources.

Response 2: Authors agree that Results section should contain obtained reults of experiments and Discussion section should  embrance deliberation on results. Phrase „ and discussion” was delated from Page 2; line 97.

Response 3: The sentence „The preliminary HPLC-DAD analysis of the Seseli devenyense methanolic fruit extract (SDMFE) revealed the presence of several interesting coumarin compounds (Figure S1)” was changed to „The chromatogram obtained as results of preliminary HPLC-DAD analysis of the Seseli devenyense methanolic fruit extract (SDMFE) displayced the presence of severalcoumarin compounds. Their chemical character and structure were confirmed by HPLC-ESI-QTOF-MS/MS(Figure S1)”.

Response 4: Figure 1 and Figure were moved to Supplementary material- Part A.

Response 5:Literature source of spectral data for d-Laserpitin, Isolaserpitin and cis-khellactone were supplemented. References at position 19 and 20.

Response 6: Page 5; line 146; “from Widelski et al. (2005)” was corrected to „Widelski et al. ”

Response 7: References for d-Laserpitin and Isolaserpitin were added in position 19.

Response 8: Figure 4-9 were corrected as indicated  by the Editor.